# Friend or Foe: Symbiotic Bacteria in *Bactrocera dorsalis*–Parasitoid Associations

**DOI:** 10.3390/biology12020274

**Published:** 2023-02-09

**Authors:** Rehemah Gwokyalya, Christopher W. Weldon, Jeremy Keith Herren, Joseph Gichuhi, Edward Edmond Makhulu, Shepard Ndlela, Samira Abuelgasim Mohamed

**Affiliations:** 1International Centre of Insect Physiology and Ecology, Nairobi P.O. Box 30772-00100, Kenya; 2Department of Zoology and Entomology, University of Pretoria, Private Bag X20, Pretoria 0028, South Africa

**Keywords:** *Diachasmimorpha longicaudata*, *Lactococcus lactis*, *Citrobacter freundii*, *Fopius arisanus*, Tephritidae, fruit fly, biological control, gut symbionts, fitness

## Abstract

**Simple Summary:**

Host-associated gut bacteria influence the eco-physiological functions of their hosts, for example, how they interact with natural enemies in their ecosystems, and hence the outcome of biological pest management regimes. Here, we assessed the effect of three common gut bacteria of the oriental fruit fly, *Bactrocera dorsalis*, on the biological control of this pest using parasitoid wasps. We found that some gut bacteria increased parasitoid emergence and positively impacted the size and fecundity of the emerging parasitoid offspring. We, therefore, conclude that some bacteria can be used as probiotics in the mass rearing of parasitoids to boost the biological control of the oriental fruit fly.

**Abstract:**

Parasitoids are promising biocontrol agents of the devastating fruit fly, *Bactrocera dorsalis*. However, parasitoid performance is a function of several factors, including host-associated symbiotic bacteria. *Providencia alcalifaciens*, *Citrobacter freundii*, and *Lactococcus lactis* are among the symbiotic bacteria commonly associated with *B. dorsalis*, and they influence the eco-physiological functioning of this pest. However, whether these bacteria influence the interaction between this pest and its parasitoids is unknown. This study sought to elucidate the nature of the interaction of the parasitoids, *Fopius arisanus*, *Diachasmimorpha longicaudata*, and *Psyttlia cosyrae* with *B. dorsalis* as mediated by symbiotic bacteria. Three types of fly lines were used: axenic, symbiotic, and bacteria-mono-associated (*Lactococcus lactis*, *Providencia alcalifaciens*, and *Citrobacter freundii*). The suitable stages of each fly line were exposed to the respective parasitoid species and reared until the emergence of adult flies/parasitoids. Thereafter, data on the emergence and parasitoid fitness traits were recorded. No wasps emerged from the fly lines exposed to *P. cosyrae*. The highest emergence of *F*. *arisanus* and *D. longicaudata* was recorded in the *L. lactis* fly lines. The parasitoid progeny from the *L. lactis* and *P. alcalifaciens* fly lines had the longest developmental time and the largest body size. Conversely, parasitoid fecundity was significantly lower in the *L. lactis* lines, whereas the *P. alcalifaciens* lines significantly improved fecundity. These results elucidate some effects of bacterial symbionts on host–parasitoid interactions and their potential in enhancing parasitoid-oriented management strategies against *B. dorsalis*.

## 1. Introduction

The oriental fruit fly, *Bactrocera dorsalis* (Hendel) (Diptera: Tephritidae), is a major horticultural pest that infests numerous fruit and vegetable crops across Asia, Africa, Oceania, and the Americas, resulting in hefty losses to the horticultural sectors of affected countries [1,2,3,4]. The management of this invasive pest is largely dependent on chemical insecticides, with far-reaching consequences on human and environmental health [5]. Furthermore, the overreliance on these chemicals very often results in the development of pest resistance. To ameliorate these effects, integrated pest management (IPM) is slowly being embraced in various parts of the world [6,7,8]. One of the IPM tactics being deployed is the use of biological control agents such as parasitoids [9,10,11,12], which, when released in sufficient numbers under ideal conditions, can significantly suppress pest populations [11,12,13]. 

The egg-prepupal parasitoid *Fopius arisanus* (Sonan) (Hymenoptera: Braconidae) and the larval-prepupal parasitoid *Diachasmimorpha longicaudata* (Ashmead) (Hymenoptera: Braconidae) are co-evolved, natural enemies of *B. dorsalis* [14,15,16]. These parasitoids have been credited with the most outstanding successful biological control programs against tephritids in Hawaii [11,12] and French Polynesia [13,17]. *Fopius arisanus* and *D. longicaudata* were imported into Africa for the classical biological control of *B. dorsalis* [9,18,19], especially because the pest lacked efficient resident parasitoid species on the continent [20]. One of the native parasitoid species of Africa, *Psyttalia cosyrae* (Wilkinson) (Hymenoptera: Braconidae), has also been investigated for possible use in the management of this invasive pest alongside *D. longicaudata* and *F. arisanus* [8].

Previous research has shown great variations in parasitism success by these three parasitoids on *B. dorsalis*. Successful development has been recorded for *D. longicaudata* and *F. arisanus* [14,16,18,21] but none for *P. cosyrae* [15,22]. While these variations in parasitism success could be due to the ability of the parasitoids to surmount their hosts’ defense mechanisms [22], there has been an increasing appreciation of other factors such as host-associated symbionts and their roles in shaping the dynamics of host–parasitoid interactions [23,24,25,26,27,28].

Bacterial symbionts have been shown to alter the outcomes of host–parasitoid interactions, with the majority of the bacteria showing host-protective functions [23,26,29,30,31,32,33]. These symbiont-conferred protective phenotypes have been linked to the symbionts’ ability to synthesize endotoxins [34,35,36] or alter host immune and metabolic processes [37,38]. As a result, the development of the parasitoid offspring is hampered. In addition to their protective roles, symbiotic bacteria have been shown to affect host selection and oviposition cues by parasitoids, as well as the fitness parameters of parasitoid progenies, by influencing their developmental times, sex ratios, size, and survival [39,40,41,42,43,44]. The ability of bacterial symbionts to alter the dynamics of host–parasitoid interactions is significant for the ecological dynamics of the host insects and parasitoids. This relationship could be exploited to develop symbiont-based biocontrol management strategies for pests of agricultural importance, such as the invasive *B. dorsalis*. 

Diverse communities of symbiotic bacteria are associated with *B. dorsalis* [4,45,46]. Some of these bacteria have been shown to alter the development, foraging behavior, mating, and immunity of their host [46,47,48,49,50]. Specifically, *Lactococcus lactis*, *Providencia alcalifaciens*, and *Citrobacter freundii* are among the most common gut bacteria associated with *B. dorsalis* [46,51,52]. These bacteria have been implicated in the detoxification of insecticides, performance of irradiated flies, host fly fitness, and susceptibility to entomopathogenic fungi [5,46,51]. Therefore, microbial symbionts are key mediators of the physiological and ecological roles of *B. dorsalis*. However, whether they affect the outcome of the interactions between *B. dorsalis* and its associated parasitoids has not been fully elucidated. 

Since such bacterial symbionts present considerable physiological and ecological benefits to their host, *B. dorsalis*, it is of interest to determine whether they promote host-protective phenotypes against parasitic wasps and if they influence the fitness parameters of the parasitoid progeny. Hence, the current study aimed at determining whether *L. lactis*, *P. alcalifaciens*, and *C. freundii*, the symbionts commonly associated with *B. dorsalis*, (1) affect parasitoid oviposition behaviors, (2) confer general protection against the braconid parasitoids associated with *B. dorsalis*, and (3) alter the fitness parameters of *D. longicaudata*, *F. arisanus*, and *P. cosyrae* progeny.

## 2. Materials and Methods

### 2.1. Bactrocera dorsalis and Parasitoid Rearing

The *Bactrocera dorsalis* flies used in this study were reared on ripe, insecticide-free mangoes, as previously described [22], in the insectary of the International Centre of Insect Physiology and Ecology (*icipe*). The flies were derived from the eggs collected from a 15th-generation, laboratory-reared colony, which was initiated with field collections and was regularly infused with field collections and maintained, as previously described [53]. The eclosed adult flies were fed on yeast and water ad libitum [15]. The parasitoids used in this study, *F. arisanus*, *D. longicaudata*, and *P. cosyrae*, were also reared in the insectary at *icipe* at a temperature range of 25–27 °C, 60–70% relative humidity, and a 12:12 day:light photoperiod. *Psyttalia cosyrae* was reared on a laboratory colony of *Ceratitis cosyra* Walker (Diptera: Tephritidae) [54], while *D. longicaudata* and *F. arisanus* were reared on *B. dorsalis* as described by Mohamed et al. [9] and Mohamed et al. [18], respectively. The parasitoids *D. longicaudata*, *F*. *arisanus*, and *P*. *cosyrae* used in this study were at the 183rd, 164th, and 177th generations in the laboratory, respectively. 

### 2.2. Bacteria Culturing

The bacteria *P. alcalifaciens*, *L. lactis*, and *C. freundii* used in this study were retrieved from glycerol stocks (stored at −80 °C) previously isolated from *B. dorsalis* [46]. Using a sterile tip, individual species, each 5 μL, were retrieved from the glycerol stocks, spread on a nutrient-rich medium, i.e., brain heart infusion (BHI) agar (Thermo Scientific™ Oxoid, Hampshire, UK), and incubated at 37 °C for 14 h. They were then re-cultured in BHI broth (Thermo Scientific™ Oxoid, Hampshire, UK) at 37 °C for 14 h in a shaking incubator.

The bacteria were harvested by centrifuging at 12879 RCF for five minutes, and the precipitate containing the bacteria was diluted in distilled water to 1 × 10^8^ concentration.

### 2.3. Generation of Axenic, Symbiotic, and Mono-Associated Bacterial Lines

The generation of bacteria-free fly lines was performed as described by Gichuhi et al. [46]. Briefly, freshly laid *B. dorsalis* eggs were collected from perforated mango domes (used as oviposition sites), surface-sterilized in 70% ethanol, dechorionated in a 7% *v*/*v* sodium hypochlorite solution for three minutes and rinsed in distilled water thrice. To confirm the elimination of bacteria from the flies, the dechorionated embryos were inoculated in autoclaved liquid larval diets [55] (modified by excluding streptomycin and nipagen) and left to develop until the 2nd instar. Consequently, 10 2nd instar larvae were randomly selected and individually homogenized in 100 µL of phosphate-buffered saline (PBS), and the homogenate was plated on BHI agar plates. The plates were incubated at 37 °C for 14 h and observed for bacterial growth. There was no bacterial growth observed on the plates, and this confirmed the axenic state of the larvae.

From these dechorionated embryos, negative (axenic) and positive (symbiotic) controls, as well as the bacterial mono-associated lines (BMALs) (*L. lactis* or *P. alcalifaciens* or *C. freundii*), were generated. Axenic lines were generated by inoculating the dechorionated embryos on autoclaved artificial liquid larval diets [55] (modified by excluding streptomycin and nipagen) supplemented with distilled water only. The BMALs were generated by directly inoculating the dechorionated eggs on autoclaved artificial liquid larval diets [55] (modified by excluding streptomycin and nipagen) supplemented with 1000 μL of 1 × 10^8^ CFU/mL inoculum of the respective bacterial isolates, individually. Symbiotic fly lines were generated by directly inoculating the dechorionated embryos on non-autoclaved larval diets previously fed upon by larvae with an intact microbiome, i.e., from a non-dechorionated embryo fly line reared under normal laboratory conditions. This unsterilized/recycled diet was selected because it would contain microbiota from the frass of the previously reared larvae. For each type of fly line, inoculations per replicate were carried out as previously described [46], with a slight modification in the size of tubes used: flat-based glass tube cylinders (75 × 25 mm). The tubes were capped with autoclaved cotton wool and maintained in a microbiological safety cabinet) at 25–27 °C, 60–70% relative humidity, and 12:12 day:light photoperiod. To minimize confounding effects, all fly lines were generated from the same batch of collected embryos that were dechorionated in the same batch before the respective inoculations and then reared under the exact same conditions.

### 2.4. Parasitic Wasp Infection of BMALs and Axenic and Symbiotic Fly Lines 

For *D. longicaudata* and *P. cosyrae* parasitization assays, *B. dorsalis* larvae were used. Briefly, *B. dorsalis* embryos were dechorionated and raised as described above for each experimental group until the larvae reached the 2nd instar stage. Using soft forceps, a set of a hundred 2nd instar *B. dorsalis* larvae were randomly selected from each fly line and individually transferred to larval oviposition units containing autoclaved carrot diets [56] (modified by excluding nipagen). The carrot diets for the BMALs and axenic and symbiotic lines were supplemented with 500 μL (1 × 10^8^) of the respective bacterial species, distilled water, or a liquid diet previously fed on by larvae with an intact microbiome, respectively. The oviposition units were transferred to a Perspex cage (12 × 12 × 12 cm) holding 10 *D. longicaudata* or *P. cosyrae* mated female parasitoids for parasitization. In a recent study, Gwokyalya et al. [22] demonstrated that *D. longicaudata* achieved oviposition above 90%, whereas *P. cosyrae* achieved similar oviposition rates when host exposure was carried out for six hours. Consequently, in this study, exposure to *D. longicaudata* was implemented for two hours, while exposure to *P. cosyrae* was carried out for six hours. Post-parasitization, the host larvae were retrieved from the oviposition units and transferred to the autoclaved carrot diets. The carrot diets were supplemented with 500 µL of the respective bacteria inoculum for the BMALs, a regular liquid larval diet for the symbiotic lines, or distilled water for the axenic lines. The host larvae were left to feed and subsequently pupariate. The puparia were transferred to four-liter transparent lunch boxes (18 × 11 × 15 cm) covered with a cotton mesh and monitored until fly and/or parasitoid emergence. 

*Fopius arisanus* parasitization assays were conducted using *B. dorsalis* eggs. Briefly, 150 dechorionated embryos were transferred to a filter paper laid on an agar plate and moistened with a 1000 µL (1 × 10^8^) inoculum of the respective bacteria species for each BMAL. For the axenic lines, distilled water was used to moisten the filter paper, whereas, for the symbiotic lines, 1000 µL of the liquid diet used to raise normal laboratory flies was used. The agar plates were incubated for one hour (to allow for efficient bacterial infection in mono-associated and symbiotic lines), after which the oviposition units were transferred to individual Perspex cages (12 × 12 × 12 cm) containing 10 mated *F. arisanus* females for parasitization, for six hours. Post-parasitization, *B. dorsalis* eggs were retrieved from the oviposition units and transferred to autoclaved liquid larval diets [55] (excluding streptomycin and nipagen) supplemented with 500 µL of the respective bacteria inoculum for the BMALs, a normal larval liquid diet for the symbiotic lines, or distilled water for the axenic lines and left to feed and subsequently pupariate. Consequently, the puparia were transferred to transparent 4-liter lunch boxes (18 × 11 × 15 cm) covered with a cotton mesh and monitored for fly and/or parasitoid emergence.

The parasitization experiments for each fly line and parasitoid were conducted individually in sterile conditions. All the wasps used in these assays underwent parasitization by exposing them to bacteria-free *B. dorsalis* larvae or eggs prior to parasitization assays. The wasps that did not exhibit oviposition behavior within 15 min were removed and replaced with new ones. All fly lines and parasitoids were kept at 25–27 °C, 60–70% humidity, and 12:12 day:light photoperiod.

### 2.5. Effect of Bacterial Symbionts on Host Acceptability for Oviposition by Parasitoid Females

To check for possible variations in acceptability by parasitoids across the different fly lines, host larvae and/or eggs from all fly lines were exposed to parasitoids, as described in Section 2.4, in no-choice assays and dissected to check for oviposition success. *Diachasmimorpha longicaudata-* and *P. cosyrae*-parasitized fly lines were dissected six hours post-parasitization, while those exposed to *F. arisanus* were dissected after four days. The host larvae containing parasitoid egg(s)/larva(e) were recorded and expressed as a percentage of the total number of exposed hosts to reflect host acceptability. These experiments were replicated five times. 

### 2.6. Parasitoid Emergence and Fitness Assays

To investigate the effect of bacterial symbionts on parasitism success and fitness of the emerging parasitoids, the host fly lines were generated and parasitized, as described in Section 2.4. The parasitized fly lines were reared until pupation, and the pupae were collected, transferred to transparent four-liter lunch boxes (18 × 11 × 15 cm) covered with a cotton net cloth, and monitored for host fly and parasitoid emergence. The emerging host flies were collected, counted, and killed by freezing, while the parasitoids were transferred to separate transparent four-liter lunch boxes (18 × 11 × 15 cm) and fed on 50% honey solution *ad libitum*. The emerged wasps from each line for each parasitoid species were counted and sexed, and the number of each sex was recorded. On day 28 of post-exposure to parasitoid wasps, the host cadavers (unhatched host puparia) were dissected to check for the presence of dead parasitoid offspring. If found, these were recorded as parasitoid deaths. 

The total number of emerging parasitoids was expressed as a percentage of the total recovered puparia to reflect parasitoid emergence. The total number of emerging host flies was expressed as a percentage of the total recovered puparia to reflect fly emergence. The total number of host puparia containing a dead parasitoid was expressed as a percentage of the total recovered puparia to reflect parasitoid death. Sex ratios were calculated as the proportion of female wasps out of all the emerged parasitoids.

### 2.7. Effect of Bacterial Symbionts Parasitoid Offspring’s Fitness Traits

#### 2.7.1. Parasitoid Progeny Developmental Time

In a separate set of experiments, *B. dorsalis* larvae and eggs were exposed to *D. longicaudata* and *F. arisanus* for two and six hours, respectively, and treated in a similar manner as described in Section 2.4. The host puparia were transferred to four- liter lunch boxes and monitored until the onset of parasitoid emergence upon which the number of emerging male and female parasitoids was recorded daily until the last parasitoid emerged. These data were used to calculate the developmental time of the parasitoids, which was computed using the formula below [14]. These experiments were replicated 10 times.
$$\mathrm{Developmental}\;\mathrm{time}=\frac{\sum_{i=1}^{n}nDi*nFi}{\sum_{i=1}^{n}\;nFi}$$
where i signifies an individual of a total of *n* insects; *nF*i is the daily individual insect (parasitoid) emergence; and *nD*i is the duration in days for the development of the ith insect (parasitoid) from oviposition to adult emergence.

#### 2.7.2. Parasitoid Progeny Body Size

The hind tibia of the F1 parasitoid progeny of *D. longicaudata* and *F. arisanus* were used as a proxy to measure the parasitoid progeny size. Briefly, one hind leg of each of the 10-day-old F1 parasitoids was removed, mounted on a microscope slide using entellan^TM^ rapid mounting medium for microscopy (Merck, Darmstadt, Germany), and layered with coverslips. The hind tibiae (one tibia from each parasitoid) were measured under a stereomicroscope (Zeiss Stemi 508, Zeiss, Oberkochen, Germany). From each fly line, 10 individuals (five males and five females) of each parasitoid species were randomly selected for the assays, and this was replicated five times.

#### 2.7.3. Longevity of Host-Deprived F1 Parasitoid Progeny 

From the same groups of emerging parasitoids described in Section 2.7.1 above, five replicates were randomly selected for the parasitoid progeny survival analysis. From each selected replicate of each F1 parasitoid line, 10 male and 10 female parasitoids were randomly selected, transferred to sterile transparent four-liter lunch boxes (18 × 11 × 15 cm), and fed on 50% honey solution *ad libitum*, but they were not provided with hosts for oviposition. The parasitoids were monitored daily for mortality until the last parasitoid died. For each F1 progeny line within a parasitoid species, the survival data were obtained for 50 males and 50 females.

#### 2.7.4. Fecundity of F1 Female Parasitoid Progeny

In a separate set of assays, 100 host larvae or 150 eggs from the axenic line, symbiotic line, and BMALs were exposed to 10 mated female *D. longicaudata* or *F. arisanus* for 2 and 6 hours, respectively, and incubated until parasitoid emergence, as described in Section 2.5. The emerging F1 parasitoids of each species from each fly line were used to investigate the symbiont-mediated effects on F1 parasitoid fecundity. Briefly, 10 randomly selected 5-day-old, mated females of *D. longicaudata* or *F. arisanus* from each fly line were transferred to four-liter lunch boxes (18 × 11 × 15 cm) and exposed to axenic larvae or eggs for two or six hours, respectively. The exposure was carried out once every day for 15 consecutive days. A 1:10 parasitoid: host ratio was maintained throughout the experiment. If a female parasitoid died, the number of host larvae/eggs offered daily was adjusted to maintain this ratio. The *Diachasmimorpha longicaudata*-parasitized larval groups were dissected 48 h after parasitization, while the *F. arisanus*-parasitized groups were dissected after four days, under a stereomicroscope (Zeiss Stemi 508, Zeiss, Oberkochen, Germany). The total number of parasitoid eggs and/or larvae in each host larva was recorded daily, and this was presented as the number of eggs laid per female parasitoid per day. These assays were performed in four independent replicates for each parasitoid species from each fly line. 

### 2.8. Data Analysis 

All analyses were performed using R software (version 4.0.1) [57].

The data obtained from host acceptability bioassays were analyzed using a generalized linear model (GLM) with quasi-Poisson distribution considering parasitoid species and host infection status as fixed factors. Data from the host suitability (parasitoid and fly emergence as well as parasitoid mortality) bioassay and the parasitoid progeny fitness trait (sex ratio, developmental time, body size, fecundity, and longevity) bioassay were analyzed separately for each parasitoid species. Since they were collected from the same batch of host fly lines, parasitoid and fly emergence, as well as parasitoid mortality data, were analyzed using multivariate analysis of variance (MANOVA), with host infection status as a fixed factor. In cases in which a significant effect of host infection status was detected, a univariate one-way analysis of variance (ANOVA) was conducted to check for significant differences within the response variables (parasitoid emergence, fly emergence, and parasitoid death), and this was followed by Student–Neuman–Keuls (SNK) post hoc multiple comparisons tests. 

The sex ratios of *D. longicaudata* and *F. arisanus* F1 progeny were analyzed using a GLM with a binomial error distribution, considering host infection status as a fixed factor. Parasitoid progeny developmental time was analyzed using a GLM assuming a Gamma error distribution with host infection status as a fixed factor. Parasitoid progeny body size (hind tibia length) was analyzed using a linear mixed-effect model from the ‘lme4’ package (version 1.1.27.1) [58], considering host infection status and the sex of the parasitoid progeny as fixed and random factors, respectively. A generalized, linear mixed-effect model (using the ‘glmer’ function from the ‘lme4’ package [58]), with a Gamma error distribution, was fitted to analyze the fecundity of the female parasitoid progeny of *F. arisanus* and *D. longicaudata*, considering host infection status and replicate population as fixed and random factors, respectively. A Cox proportional hazards model using the “coxme” function from the “survival” package (version 3.2.13) [59] was fitted to determine the effect of bacterial symbionts on the longevity of the F1 progeny of both parasitoid species. In the model, both host infection status and replicate population were included as fixed and random factors, respectively. The significance of all these models was determined using analysis of deviance with chi-square tests, and mean separation was determined using Tukey’s multiple comparison tests at α ≤ 0.05.

## 3. Results

### 3.1. Effect of Bacterial Symbionts on Host Acceptability for Oviposition by Female Parasitoids

The no-choice assays revealed that host acceptance by all the parasitoid species was not affected by symbiotic bacteria (χ^2^ = 0.518, df = 3, *p* = 0.775), parasitoid species (χ^2^ = 0.246, df = 2, *p* = 0.769), or the interaction between these two factors (χ^2^ = 0.343, df = 6, *p* = 0.994) (Figure 1).

### 3.2. Effect of Bacterial Symbionts on Host Suitability for Development of the Immature Stages of the Parasitoids

We recorded a significant effect of symbiotic bacteria on parasitoid emergence, fly emergence, and parasitoid mortality in the *F. arisanus*-parasitized fly lines (MANOVA: F_12,135_ = 5.913, *p* < 0.001). Further, analysis using ANOVA showed significant effects of bacterial symbionts on parasitoid emergence (F_4,45_ = 24.122, *p* < 0.001; Figure 2a) as well as host fly emergence (F_4,45_ = 33.708, *p* < 0.001; Figure 2b) but not parasitoid mortality (F_4,45_ = 0.872, *p* = 0.488; Figure 2c). The flies inoculated with *L. lactis* yielded the highest percentage of parasitoid wasps and the lowest percentage of eclosed host flies, while the reverse was true for those inoculated with *P. alcalifaciens* (Figure 2a,b).

In general, the effect of bacterial symbionts on the three response variables in the *D. longicaudata*-parasitized host fly lines (MANOVA: F_12,135_ = 2.666, *p* = 0.003) showed a similar trend to that recorded for *F. arisanus*. Inoculating *B. dorsalis* with symbiotic bacteria significantly influenced parasitoid emergence (F_4,45_ = 8.246, *p* < 0.001; Figure 3a) and fly emergence (F_4,45_ = 11.527, *p* < 0.001; Figure 3b) but not parasitoid mortality (F_4,45_ = 0.304, *p* = 0.874; Figure 3c). Overall, the *L. lactis*-infected fly lines yielded the highest percentage of parasitoids when compared with other host fly lines (Figure 3a). 

We did not record any parasitoid emergence from *P. cosyrae*-parasitized larval groups irrespective of the infection status. However, we found a significant multivariate effect of bacterial symbionts on host fly emergence and parasitoid mortality (MANOVA: F_8,90_ = 4.667, *p* < 0.001). Further analysis revealed that host fly emergence significantly varied across all the host fly lines (F_4,45_ = 13.419, *p* < 0.001; Figure 4a), with the highest percent fly emergence recorded in the symbiotic and *P. alcalifaciens* host fly lines (Figure 4a). Parasitoid mortality was not affected by symbiotic bacteria (F_4,45_ = 1.805, *p* = 0.145; Figure 4b).

### 3.3. Effect of Bacterial Symbionts on Parasitoid Offspring’s Fitness Traits

#### 3.3.1. Parasitoid Progeny Sex Ratio

The bacterial symbionts of *B. dorsalis* did not alter the sex ratios of the progeny of *F. arisanus* (ꭓ^2^ = 0.048, df = 4, *p* = 1, Figure 5a), nor those of *D. longicaudata* (ꭓ^2^ = 0.078, df = 4, *p* = 0.999, Figure 5b). However, the sex ratio was female-biased for both parasitoid species (Figure 5a, b).

#### 3.3.2. Parasitoid Progeny Developmental Time

Developmental times significantly varied among host fly lines for both parasitoid species (χ^2^ = 122.360, df = 4, *p* < 0.001; χ^2^ = 18.20, df = 4, *p* = 0.001, for *F. arisanus* and *D. longicaudata*, respectively), being the longest for the wasps emerging from *P. alcalifaciens* and *L. lactis* fly lines (Table 1). Overall, the average developmental time for *F. arisanus* was 23.56 ± 0.37 days, while it was 17.50 ± 0.21 days for *D. longicaudata*.

#### 3.3.3. Parasitoid Progeny Body Size and Longevity

Like other fitness parameters, the body size (as measured by the hind tibia length) of the F1 progeny of both parasitoid species was significantly affected by bacterial symbionts (*F. arisanus*: χ^2^ = 56.220, df = 4, *p* < 0.001; *D. longicaudata*: χ^2^ = 345.750, df = 4, *p* < 0.001). Overall, the parasitoids emerging from *L. lactis* and *P. alcalifaciens* fly lines had longer tibia than those emerging from other fly lines for both parasitoids (Figure 6). On the other hand, there were no symbiont-mediated effects on the longevity of the parasitoid progeny of both parasitoid species (χ^2^ = 0.006, df = 4, *p* = 1.000 and χ^2^ = 0.096, df = 4, *p* = 0.998 for *F. arisanus* and *D. longicaudata*, respectively) (Figure 7a,b).

#### 3.3.4. Fecundity of Female Parasitoid Progeny

The fecundity of the F1 progeny of both parasitoids was significantly affected by symbiotic bacteria (χ^2^ = 11.698, df = 4, *p* = 0.020 and χ^2^ = 60.834, df = 4, *p* < 0.001 for *F. arisanus* and *D. longicaudata*, respectively), with the parasitoid progeny emerging from the *P. alcalifaciens*-inoculated fly lines ovipositing more eggs (*F. arisanus* = 6.18 ± 0.29 and *D. longicaudata* = 6.74 ± 0.26 eggs/♀/day), whereas the least fecund parasitoid wasps were those reared from the *L. lactis*-inoculated fly lines (*F. arisanus* = 5.44 ± 0.16 and *D. longicaudata* = 5.30 ± 0.13 eggs/♀/day, Figure 8a,b).

## 4. Discussion

Augmentative biological control is a widely used approach for the management of fruit flies in many countries [11,12,13,16,20]. This approach does not only require the production of parasitoids on a large scale, but the produced parasitoid wasps also need to be of high quality in terms of fitness, to ensure their optimal performance in the field. Symbiotic bacteria could be exploited to enhance the quality of mass-reared parasitoids [41]. Indeed, symbiotic bacteria have been reported to influence the physiological quality of their hosts, inevitably influencing how they are perceived by parasitoids and their suitability for the development of parasitoid immatures and, eventually, the fitness of the emerging parasitoid progeny [39,41,42,60]. 

The comparable acceptability of the three parasitoid species (*F. arisanus*, *P. cosyrae*, and *D. longicaudata*) across the *B. dorsalis* fly lines reported in this study might be an indication that these symbiotic bacteria had no influence of the host cues used by the parasitoid females to locate their hosts. Indeed, Sochard et al. [27] reported that *Hamiltonella defensa* and *Regiella insecticola* did not alter the acceptability of *Acyrthosiphon pisum* (Harris) (Hemiptera: Aphididae) by its parasitoid, *Aphidius ervi* Haliday (Hymenoptera: Braconidae). Likewise, Xie et al. [32] showed that *Spiroplasma sp*. had no effect on the parasitism rates of *Leptopilina heterotoma* Förster (Hymenoptera: Figitidae) in *Drosophila melanogaster* Meigen (Diptera: Drosophilidae). However, the findings by Attia et al. [39] and Frago et al. [29] revealed that symbionts influence the host acceptability by parasitoid wasps, a phenotype that was attributed to the symbiont-mediated cues that influence host finding and ovipositing behavior by the parasitoids.

Nonetheless, despite the similar outcome in host acceptability, parasitoid emergence varied among the *B. dorsalis* fly lines, suggesting that bacterial symbionts had a profound effect on parasitoid development. Most facultative bacteria described to date possess protective roles where, in some instances, complete host protection against parasitoids is realized [30]. These protective phenotypes are largely attributed to the symbiont-encoded viruses that impair parasitoid development [35,36], the ability of the symbionts to alter the host immune defenses [37], or their ability to compete for nutrients, which are needed for the successful development of the parasitoids [38]. However, the high parasitism rates recorded in the *L. lactis*-inoculated fly lines in the current study suggest that *L. lactis* increases the susceptibility of *B. dorsalis* to attack by parasitoids. This susceptibility could be associated with the suppression of immune defense mechanisms of *B. dorsalis* and/or the *L. lactis*-mediated pathogenicity mechanism, previously highlighted in the *B. dorsalis*–*L. lactis*–*Metarhizium anisopliae* interaction [46]. However, this mechanism remains poorly understood. 

Surprisingly, one evaluated symbiont, *P. alcalifaciens*, was found to confer protection to *B. dorsalis* against *F. arisanus* but not *D. longicaudata*, suggesting that bacteria-mediated protection against parasitoids is not universal across host–parasitoid systems. These two parasitoids attack different stages of their host, with *F. arisanus* using the eggs and *D. longicaudata* using the larvae as hosts, and this might have contributed to the discrepancy in *P. alcalifaciens*-mediated protection. Alternatively, it is possible that *P. alcalifaciens* produces toxins or engages pathogenic mechanisms to which only *F. arisanus* is susceptible. Similar symbiont-specific host-protective phenotypes have been demonstrated in other host–symbiont–parasitoid tripartite interactions [31,40,61,62].

Similar to earlier studies [40,42], we found that symbiotic bacteria did not affect the sex ratios of the progeny of *F. arisanus* and *D. longicaudata*. It has been demonstrated that parasitoids discriminate between good- and poor-quality hosts and, consequently, allocate more female offspring to good-quality hosts [63,64,65,66]. The results obtained in the current study showed that the F1 progeny of both parasitoid species from all fly lines were female-biased. The bias may be caused by the parasitoids perceiving hosts across all fly lines as good quality, thus choosing to oviposit more females in all of them. Furthermore, the female parasitoids used in this study were newly mated and in their early stages of oviposition, which may have contributed to this sex bias. Studies have shown that mated female parasitoids are likely to lay more females than males since most eggs at this stage are diploid (fertilized) eggs that develop into female progeny [67,68]. 

Earlier studies suggested that the parasitoid progeny’s developmental time and size are a function of host size [42,69], wherein the parasitoids emerging from smaller hosts take longer to develop, especially for koinobiont parasitoid species [70,71]. Indeed, the *B. dorsalis* pupae from the *L. lactis* and *P. alcalifaciens* fly lines were reported to have higher weights than those from the other fly lines [46]. It was, therefore, expected that the parasitoid progeny from these lines would emerge earlier. However, in this study, these parasitoids emerged later than those reared on the, *C*. *freundii*, axenic, and symbiotic host fly lines with associated lower puparia weights. This phenotype could be a symbiont-mediated manipulation of host physiology such that the parasitoids developing in these lines delay development and subsequent emergence until the hosts reach the desired physiological state. Alternatively, it is known that the parasitoid rate of the consumption of its host is inversely proportional to the size of the host, which may increase the development time of the parasitoids, especially in tissue-feeding endoparasitoids [69,70]. It is, therefore, possible that the parasitoids from the *P. alcalifaciens* and *L. lactis* fly lines had longer developmental times, as they had to completely consume their relatively larger hosts prior to emergence. These findings, however, contrast with a previous report that other *Providencia* species shorten the developmental time and alter the sex ratio of *D. longicaudata* progeny [41]. Differences in bacteria strains used, inoculation strategy, and host species may account for the contrast between our findings and the findings of that study, more so since in the latter, bacterial isolates were supplemented in non-axenic diets, thus accommodating additive effects from other native bacterial symbionts in the host larvae.

Moreover, the finding that the parasitoids emerging from the *P. alcalifaciens* and *L. lactis* host fly lines were larger than those reared on the axenic, symbiotic, and *C*. *freundii* counterparts conforms with the host–parasitoid size theory, according to which larger hosts are predicted to yield larger parasitoid progeny [69,70,71,72,73]. The larger body size of the parasitoid progeny from these lines could, therefore, be an additive effect of these two symbionts on the size of their hosts, a trait that could be attributed to symbiont-mediated effects on the nutritional resources of the hosts reared on these bacterial lines. 

Previous studies suggested that the size of a female parasitoid is directly proportional to its fecundity and longevity, e.g., [72,74,75]. While this held true for the F1 parasitoid progeny emerging from the *P. alcalifaciens* fly lines, the opposite was recorded for those reared on the *L. lactis* fly lines, which, despite being larger than the parasitoids from the axenic, symbiotic, and *C*. *freundii* fly lines, were less fecund. This finding suggests the *L. lactis*-associated fitness costs in such a way that the female parasitoid progeny from this fly line allocate more resources towards growth and adult maintenance at the detriment of their reproductive potential. This symbiont may, therefore, have cryptic fitness costs on the reproductive potential of parasitoids that supersede the progeny size benefits. Similar symbiont-mediated tradeoffs in parasitoid fitness have been demonstrated in other host–symbiont–parasitoid systems [39,62,76].

## 5. Conclusions

This study reveals a mixed nature of interactions: a general synergism with the parasitoids with *L. lactis* and a more beneficial role to the host flies in the case of *P. alcalifaciens*, suggesting that symbiotic-bacteria could have significant consequences on host–parasitoid dynamics. In this regard, these findings underscore the previously unknown role of bacterial symbionts in mediating the host–parasitoid interactions in *B. dorsalis* by providing information vital for parasitoid mass rearing. For example, *L. lactis* can be harnessed as a probiotic in mass-rearing programs of *D. longicaudata* and *F. arisanus*, hence enhancing the performance of parasitoids in biological control programs targeting this pest. Nevertheless, future studies are needed to elucidate the mechanisms underlying these symbiont-mediated host–parasitoid interactions.

## Figures and Tables

**Figure 1 biology-12-00274-f001:**
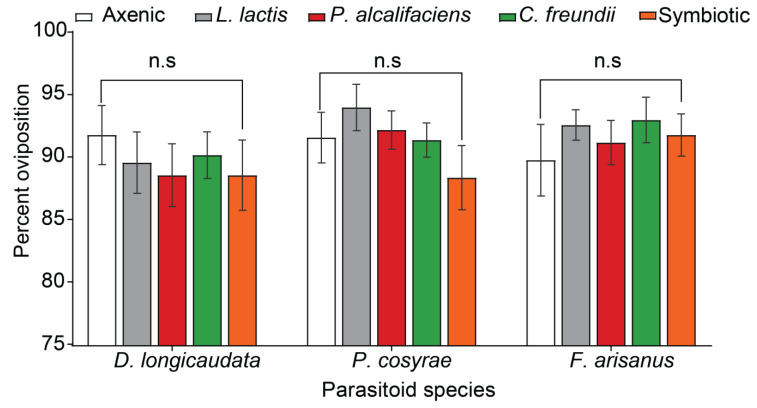
Acceptability by three parasitoid species (*Diachasmimorpha longicaudata*, *Fopius arisanus*, and *Psyttalia cosyrae*) for *Bactrocera dorsalis* fly lines (axenic, *Lactococcus lactis*, *Providencia alcalifaciens*, *Citrobacter freundii*, and symbiotic) (mean ± standard error of the mean (SEM)). n.s denotes non-significant differences (Tukey’s tests, α ≤ 0.05, *n* = 500).

**Figure 2 biology-12-00274-f002:**
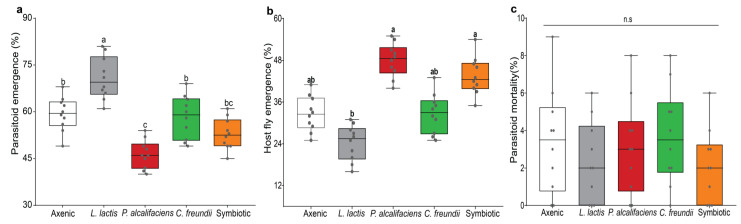
Effects of symbiotic bacteria (axenic, *Lactococcus lactis*, *Providencia alcalifaciens*, *Citrobacter freundii*, and symbiotic) on the outcome of *Bactrocera dorsalis* and *Fopius arisanus* interaction. Within each category ((**a**) parasitoids, (**b**) host flies, and (**c**) parasitoid mortality), boxes capped with different letters indicate significant statistical differences; n.s denotes nonsignificant differences (Student–Neuman–Keuls (SNK) test, α ≤ 0.05, *n* = 1000). Within each box, horizontal bars denote the median values, and the ends of each boxplot whisker represent the minimum and maximum values of the data. Each data point represents the percent individuals (emerging parasitoids and host flies as well as dead parasitoids) recorded for a single replicate.

**Figure 3 biology-12-00274-f003:**
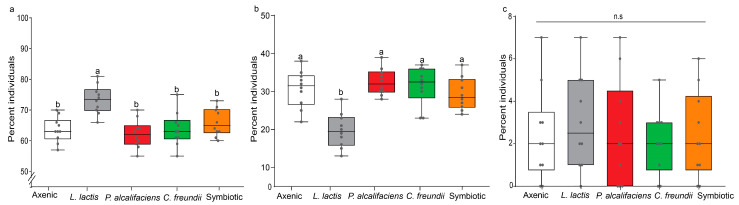
Effects of symbiotic bacteria (axenic, *Lactococcus lactis*, *Providencia alcalifaciens*, *Citrobacter freundii*, and symbiotic) on the outcome of *Bactrocera dorsalis* and *Diachasmimorpha longicaudata* interaction. Within each category ((**a**) parasitoids, (**b**) host flies, and (**c**) parasitoid mortality), boxes capped with different letters indicate significant statistical differences; n.s denotes nonsignificant differences (Student–Neuman–Keuls (SNK) test, α ≤ 0.05, *n* = 1000). Within each box, horizontal bars denote the median values and the ends of each boxplot whisker represent the minimum and maximum values of the data. Each data point represents the percent individuals (emerging parasitoids and host flies as well as dead parasitoids) recorded for a single replicate.

**Figure 4 biology-12-00274-f004:**
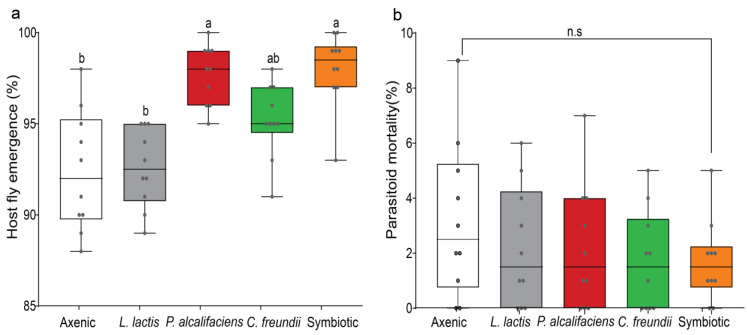
Effects of symbiotic bacteria (axenic, *Lactococcus lactis*, *Providencia alcalifaciens*, *Citrobacter freundii*, and symbiotic) on the outcome of *Bactrocera dorsalis* and *Psyttalia cosyrae* interaction. Within each category ((**a**) host flies and (**b**) parasitoid mortality), boxes capped with different letters indicate significant statistical differences; n.s denotes nonsignificant differences (Student–Neuman–Keuls (SNK) test, α ≤ 0.05, *n* = 1000). Within each box, horizontal bars denote the median values and the ends of each boxplot whisker represent the minimum and maximum values of the data. Each data point represents the percent individuals (emerging host flies as well as dead parasitoids) recorded for a single replicate.

**Figure 5 biology-12-00274-f005:**
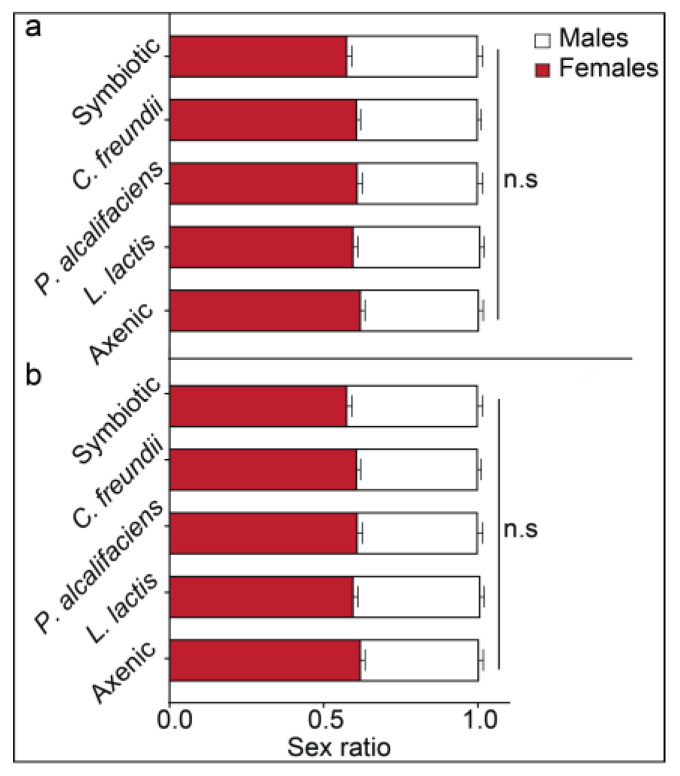
Sex ratios (mean ± SEM) of (**a**) *Fopius arisanus* and (**b**) *Diachasmimorpha longicaudata* progeny emerging from different *Bactrocera dorsalis* lines (axenic, *Lactococcus lactis*, *Providencia alcalifaciens*, *Citrobacter freundii*, and symbiotic). n.s denotes non-significant differences among the sex ratios of the emerging parasitoids (Tukey’s tests, α = 0.05).

**Figure 6 biology-12-00274-f006:**
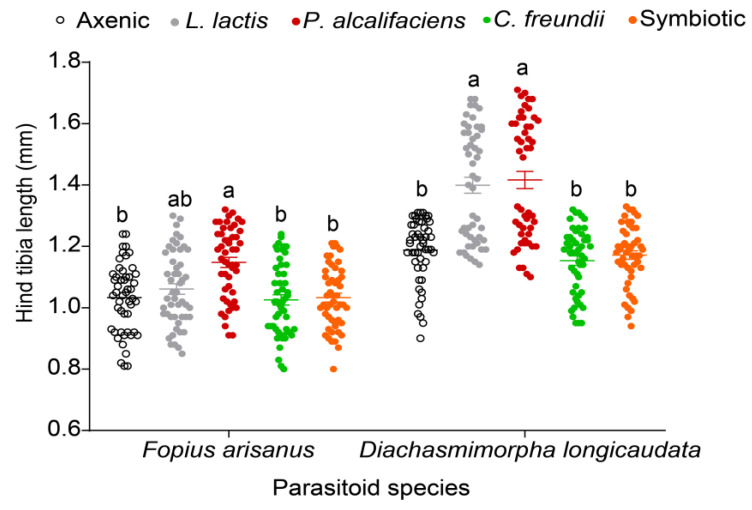
Effect of *Bactrocera dorsalis* symbiotic bacteria (axenic, *Lactococcus lactis*, *Providencia alcalifaciens*, *Citrobacter freundii*, and symbiotic) on F1 parasitoid body size (hind tibia lengths (mm; mean ± standard error of the mean (SEM)). Within each parasitoid species, scatter plots capped with different letters are significantly different (Tukey’s tests, α = 0.05, *n* = 50). Each data point represents the tibia length of a single adult parasitoid.

**Figure 7 biology-12-00274-f007:**
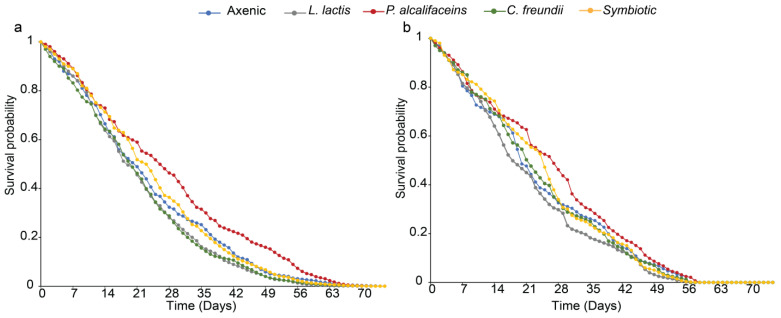
Effect of *Bactrocera dorsalis* symbiotic bacteria (axenic, *Lactococcus lactis*, *Providencia alcalifaciens*, *Citrobacter freundii*, and symbiotic) on the survival of F1 progeny of (**a**) *Fopius arisanus* and (**b**) *Diachasmimorpha longicaudata* (Tukey’s tests, α = 0.05, *n* = 100). The data represent 100 (50 males and 50 females) *F. arisanus* and *D. longicaudata* F1 offspring kept in four-liter lunch boxes and fed with 50% honey solution ad libitum. Survival of both parasitoid species was recorded until the last parasitoid died.

**Figure 8 biology-12-00274-f008:**
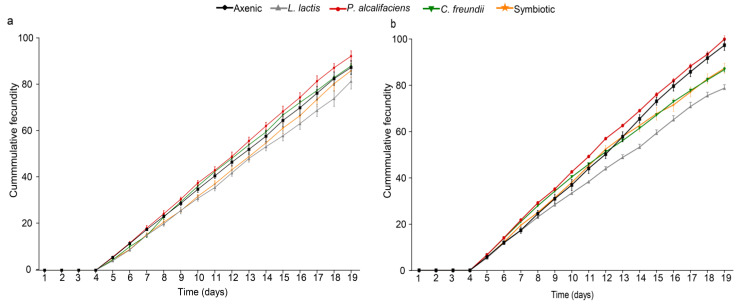
Cumulative fecundity of F1 female parasitoid progeny of (**a**) *Fopius arisanus* and (**b**) *Diachasmimorpha* longicaudata reared on different *Bactrocera dorsalis* fly lines (axenic, *Lactococcus lactis*, *Providencia alcalifaciens*, *Citrobacter freundii*, and symbiotic) (Tukey’s tests, α = 0.05, *n* = 400). Error bars reflect the standard error of the mean (SEM).

**Table 1 biology-12-00274-t001:** Effect of bacterial symbionts on the developmental times of *Fopius arisanus* and *Diachasmimorpha longicaudata* progeny emerging from the different *Bactrocera dorsalis* fly lines (axenic, *Lactococcus lactis*, *Providencia alcalifaciens*, *Citrobacter freundii*, and symbiotic). Within each column, means followed by the same letter are not significantly different (Tukey’s tests, α = 0.05, *n* = 1000). SEM = standard error of the mean.

Developmental Time Days (Mean ± SEM)				
Parasitoid Species	Fly Line	Male	Female	Pooled
*Fopius arisanus*	Axenic	19.96 ± 0.109 a	21.39 ± 0.118 a	20.76 ± 0.133 ab
	*L. lactis*	20.30 ± 0.147 ab	21.86 ± 0.083 b	21.05 ± 0.174 bc
	*P. alcalifaciens*	20.69 ± 0.077 b	21.80 ± 0.074 b	21.16 ± 0.210 c
	*C. freundii*	19.99 ± 0.115 a	21.29 ± 0.122 a	20.78 ± 0.086 ab
	Symbiotic	19.93 ± 0.078 a	21.12 ± 0.084 a	20.55 ± 0.118 a
*Diachasmimorpha longicaudata*	Axenic	14.97 ± 0.977 a	17.65 ± 0.096 a	16.97 ± 0.107 a
	*L. lactis*	16.91 ± 0.220 b	18.57 ± 0.243 c	17.89 ± 0.224 b
	*P. alcalifaciens*	16.75 ± 0.101 ab	18.18 ± 0.048 bc	17.61 ± 0.052 b
	*C. freundii*	15.81 ± 0.113 ab	17.56 ± 0.096 a	16.81 ± 0.106 a
	Symbiotic	16.08 ± 0.117 ab	17.72 ± 0.097 ab	16.99 ± 0.096 a

## Data Availability

The datasets generated during and/or analyzed during the current study are available from the corresponding author(s) on reasonable request.

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
