# Peer review of "Friend or Foe: Symbiotic Bacteria in Bactrocera dorsalis–Parasitoid Associations"

_biology, 2023, doi:10.3390/biology12020274_

Round 1

Reviewer 1 Report

Title

The title is concise and informative but still needs improvement to produce a good sound.

Keywords

Please add some strong key words

Abstract

Complete

Introduction

Complete

Materials and methods

This portion provided a sufficient detailed methodology followed.

Results and Discussion

Results are clear concise and well presented. Some grammatical mistakes are present that need improvement.

References 
Please ensure that every reference cited in the text is also present in the reference list. Follow the general guideline of the Journal for references.

Reviewer 2 Report

November 28, 2022

Journal of Biology

 Dear Authors,

Attached you will find my comments and suggestions about the manuscript “Friend or foe: Symbiotic bacteria in Bactrocera dorsalis-parasitoid associations”. Written by Rehemah Gwokyalya et al. (biology-2054558-v1).

General comments

P3, L98, 102 and 105.- Include from wich generation was each mass-reared specimen.

P5, L227-228.- How many tiae were measured? Please, explain.

P5, L229-230.- Describe in more detail the number of treatments, repetitions and total of experimental units considered in this test.

P6; 255-256.- Describe in more detail the number of treatments, repetitions and total of experimental units considered in this test.

P6; L258.- Move the paragraph at the end of the data analysis section.

P7, L295-296, 303, 306, 311-312.- Write all scientific names in italics.

P11; L391-395, 398-399.- Write all scientific names in italics.

P8-L318-319, 323, 326-327, 333 and 338.- Write all scientific names in italics.

P13; L484.- Change “Empirical studies,” by “Previous reports,”

P14; L574.- Delete the words what is indicated.

Reviewer 3 Report

Dear Author.

I am interested in reading this article because the study is interesting and has a novelty value, which is well written in the abstract and introduction, clearly and supported by adequate literature. I also read that the methodology including the qualifications of the materials and tools used to explore the effect of the symbiont bacteria on the body of the host fly is very suitable, with a very detailed explanation. The research information presented in point 2 is also very good and clear. Likewise, the discussion rubric has been written in great detail with the support of suitable and up-to-date references. However, there are some things that need explanation and clarification (just to confirm), namely:

lines 224-225;  323 - 324; 437 - 444; 499 - 500 please look at the manuscript.

Reviewer 4 Report

The study presented by Gwokyalya et al., intends to determine if symbiont bacteria in the gut provide protection to flies against parasitoids, alter oviposition and the fitness of natural enemies.

In general, the manuscript is original and novel, I only have one questions for the authors and some minor recommendations.

Q1. In the discussion you mention that the B. dorsalis pupae of the L. lactis and P. alcalifaciens fly lines had higher weights than those of the other fly lines. What do you attribute this larger size to? Deepen your discussion about it.

Recommendations

L41 update references on the economic importance of B. dorsalis, the ones you present date from 19 years ago.

L46 update reference 11; Reference 10 corresponds to the memories of a symposium. Is the article currently published? If so, replace the reference

L48 inserts a current reference

L56 the word "native" appears 2 times on the same line

L218 adapts the reference to the format of the journal

L259-L287 specify the version of each R package

L276 cite the reference of each R package
